# Inequality is rising where social network segregation interacts with urban topology

Gergő Tóth[1,2,13], Johannes Wachs [3,4,13], Riccardo Di Clemente [5,6], Ákos Jakobi[7,8], Bence Ságvári[1,9,10], János Kertész [11] & Balázs Lengyel [1,10,12 ✉]

Social networks amplify inequalities by fundamental mechanisms of social tie formation such as homophily and triadic closure. These forces sharpen social segregation, which is reflected in fragmented social network structure. Geographical impediments such as distance and physical or administrative boundaries also reinforce social segregation. Yet, less is known about the joint relationships between social network structure, urban geography, and inequality. In this paper we analyze an online social network and find that the fragmentation of social networks is significantly higher in towns in which residential neighborhoods are divided by physical barriers such as rivers and railroads. Towns in which neighborhoods are relatively distant from the center of town and amenities are spatially concentrated are also more socially segregated. Using a two-stage model, we show that these urban geography features have significant relationships with income inequality via social network fragmentation. In other words, the geographic features of a place can compound economic inequalities via social networks.

[1] Agglomeration and Social Networks Lendület Research Group, Centre for Economic-and Regional Studies, Budapest, Hungary. [2] Spatial Dynamics Lab, University College Dublin, Dublin, Ireland. [3] Institute for Data, Process and Knowledge Management, Vienna University of Economics and Business, Vienna, Austria. [4] Complexity Science Hub Vienna, Vienna, Austria. [5] Department of Computer Science, University of Exeter, Exeter, UK. [6] Centre for Advanced Spatial Analysis, University College London, London, UK. [7] Department of Regional Science, Eötvös Loránd University, Budapest, Hungary. [8] Institute of Advanced Studies, Kőszeg, Hungary. [9] CSS-Recens, Centre for Social Sciences, Budapest, Hungary. [10] International Business School Budapest, Budapest, Hungary. [11] Department of Network and Data Science, Central European University, Budapest, Hungary. [12] NETI Lab, Corvinus Institute for Advanced Studies, Budapest Corvinus University, Budapest, Hungary. [13]These authors contributed equally: Gergő Tóth, Johannes Wachs. ✉email: lengyel.balazs@krtk.hu

Wealth and income inequalities are growing[1], slowing development, economic growth, and technological progress[2–4] while fostering radicalization and the advance of political populism[5,6]. These disparities have historical roots; unequal access to education, technology, and public services are self-reinforcing mechanisms by which economic inequality compounds across generations[7,8]. Less is known about how structural factors that influence inequality interact with one another. Two prime examples of such factors are social networks and geography.

Research on social networks emphasizes that social relations provide individuals with essential access to economic opportunities[9]. Social networks are claimed to maintain and even amplify inequalities when economic status plays a role in how social relations are established[10,11]. For example, a major micro-level mechanism for social-tie formation is homophily, the tendency for similar individuals to become friends[12]. Triadic closure, the phenomenon that friends of friends are more likely become friends[13], compounds the effect of homophily in tie formation[14]. Since wealth is the one of the most significant boundaries to social relations in most societies[15], these micro-level mechanisms can result in social segregation at the macro scale: groups with different socioeconomic status are separated from each other in social networks[16]. This kind of macro-scale network topology can lead to divergence of economic potentials between groups if access to resources or information runs through the network[11].

Social networks are themselves embedded in geography, which itself has fundamental connections with inequality. The location of an individual's home, for example, predicts a large share of their future economic outcomes[17]. A consequent divergence of outcomes across neighborhoods is observed even within relatively small geographical units such as cities and towns[18,19]. The local bias of social ties, by which individuals are more likely to connect with those who are close to them[20,21], and the observation that social connections are less frequent across physical or administrative boundaries[22–24], together suggest that the primary way geography influences economic outcomes is through its effects on social network structure. Indeed, spatially bounded social relations, limit both individual[25] and collective prosperity[26,27] because access to diverse resources provided by physically distant social connections is a key element for progress[28].

In this paper we study how social networks and geography interact, and together what relationship they have with inequality outcomes. Despite the rich literature relating both features with inequality directly, there has to date been little empirical work examining this interaction, likely due to the difficulty in connecting data on spatially embedded social interaction and economic outcomes. To address this gap, we analyze an online social network of roughly 2 million individuals located in about 500 towns in Hungary. We relate the degree of fragmentation in town-level social networks to income inequality, finding a vicious cycle: higher social network fragmentation compounds income inequality over time. (From here on throughout the paper we also use the term fragmentation to refer to social network fragmentation.) Next, we describe how physical urban geographic features, which are rather static and unchanging in the short run, relate to fragmentation. Using a two stage model, we show that the relationship between a town's geographic features and income inequality is mediated by social networks. We interpret our results as suggesting that when a town's geography facilitates sorting and segregation, our models predict greater inequality.

Our empirical approach rests on the observation that cities and towns are prime arenas of social interaction[29]. Previous research in city science has devoted major efforts to explain social network structure with urban characteristics, distribution of individuals, and human behavior in cities[30,31]. For example, there is

significant evidence that co-location is necessary for social ties to form and prosper[32]. Both home locations and urban activities are shown to cluster by income level, ethnic groups, and age and gender[33–36]. However, due to intensive mobility within cities[37], the emergence of spatial communities in cities have not been observed on individual level networks[38], only after aggregating connections, for example to ZIP codes in the United States[26].

It has long been theorized in economics and social sciences that the relationship between inequalities and social networks in cities are mediated by physical space[39]. For example, following Alonso[40], job-related information is shared in the city center, access to which is unequally distributed across the urban population. Social networks of the wealthy reinforce their geographic clustering around this interaction center[41]. Alternatively, self-isolation of the wealthy and powerful in peripheral neighborhoods or suburbs is another way in which social segregation manifests and economic information and opportunity is restricted[42]. In the urban sociology research, spatial barriers and boundaries are known to facilitate social segregation[43]. For instance, administrative boundaries such as school districts[44] and physical borders such as railroads or rivers[45,46] serve as landmarks that facilitate discrimination, the differential provision of public goods, and sorting. The emergent islands of segregation are difficult to bridge by social ties because poor and rich neighborhoods are typically located far from each other[47].

To better understand how the structure of built environment relates to income inequalities through social relations, we use open source geographic data and develop three measures of urban segregation of towns: (1) the average residential distance from the town center, (2) the extent of spatial concentration of amenities in towns, and (3) the degree to which physical barriers divide residential areas. Each of these indicators are significantly related to social network fragmentation. Using a machine learning approach, we find that these geographic indicators are better predictors of social network fragmentation than other social indicators of segregation.

We then deploy these indices of urban geography as instrumental variables for social network fragmentation in a regression model predicting economic inequality. Falsification tests of the instrumental variable approach support our interpretation that the geographic topology of towns has a significant relationship with economic inequality via its relationship with social network fragmentation.

## Results

**Social networks and the dynamics of inequality.** We first investigate the levels and changes of income inequality from 2011 to 2016 in all 474 Hungarian towns with at least 2500 inhabitants. We exclude capital city Budapest from the analysis because it is a unique settlement in several ways. It contains twenty administrative subunits which serve as weak social and political barriers—we do not observe this granularity in our social network data. It is also an order of magnitude outlier in population, density, and physical size and would introduce significant leverage in our regression models. The Hungarian Statistical Office provides binned data on personal income tax filings in each town in our sample including the total amount of income and the number of taxpayers in each bin (see Materials section). As an example, in Fig. 1a we compare the cumulative distribution of gross income across these bins in 2011 for a low (Ajka, in gold) and high inequality town (Gödöllő in dark blue), both having around 30,000 inhabitants.

We measure income inequality using the Gini index based on these bin distributions (see Supplementary Note 1) and then relate it to our measure of social network fragmentation at the

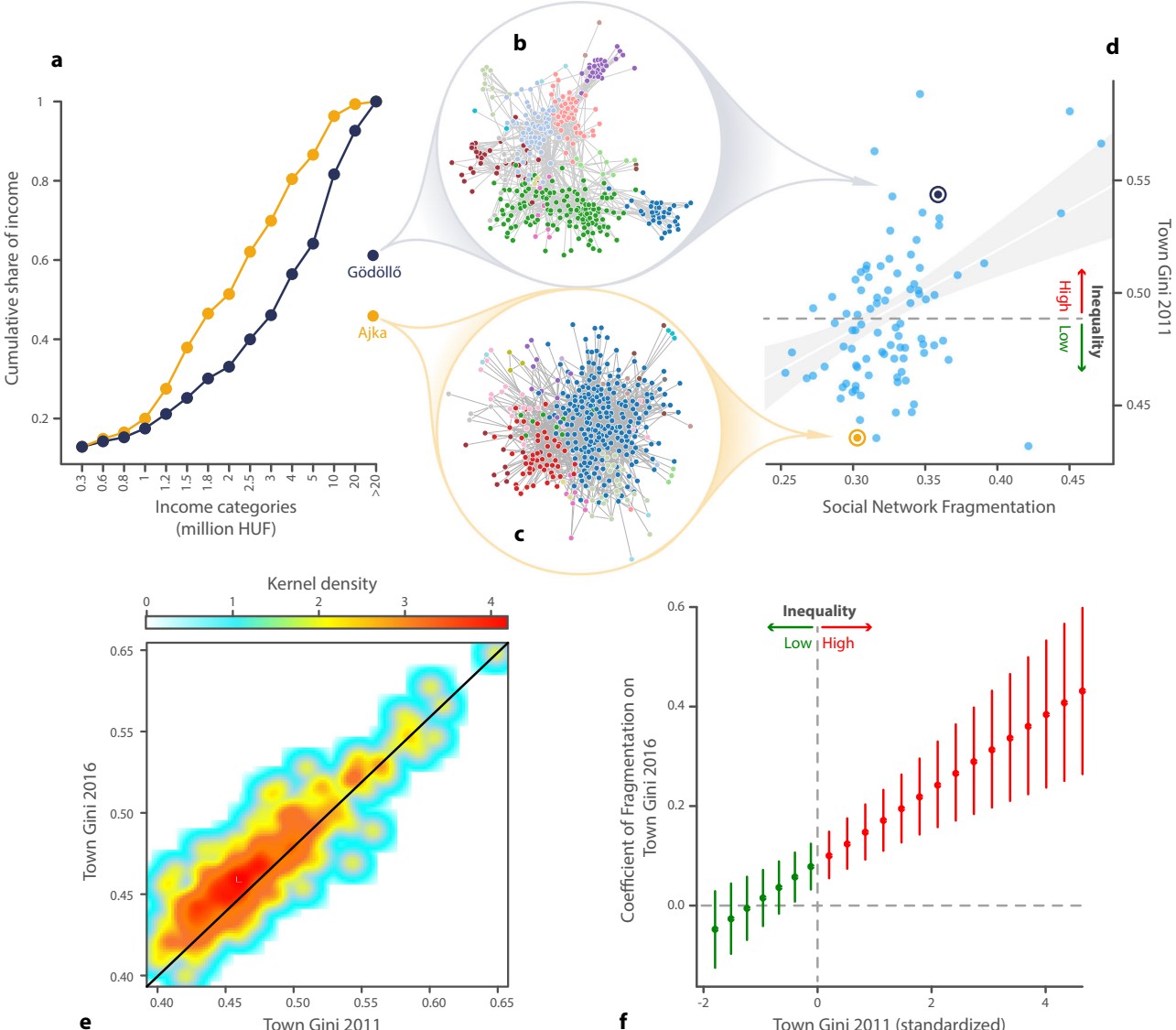

**Fig. 1 Income inequality ($G_i$) correlates with network fragmentation ($F_i$) in towns. a** Cumulative distribution of income in a relatively equal town (Ajka, gold) and a relatively unequal one (Gödöllő, dark blue). Sampled social networks in Gödöllő (**b**) and Ajka (**c**). Node colors represent network communities revealed by the Louvain method in both networks. **d** Income inequality measured by the Gini index ($G_{i,2011}$) for towns with more than 15,000 inhabitants ($n = 91$) correlates with the fragmentation ($F_i$) of their social networks (Pearson's $\rho = 0.44$). Gold dot: Ajka ($G_{i,2011} = 0.43$, $F_i = 0.3$); dark blue dot: Gödöllő ($G_{i,2011} = 0.54$, $F_i = 0.36$); blue dots: all other towns. Fitted line represents a linear regression $G_{i,2011} = 0.36 + 0.37F_i$ with $R^2 = 0.198$. The shade area depicts 95% confidence interval. The dashed horizontal line represents the mean of $G_{i,2011} = 0.488$. **e** We plot the correlation between town Gini scores in 2011 and 2016 ($G_{i,2011}$ and $G_{i,2016}$). **f** The relationship between network fragmentation ($F_i$) and inequality in 2016 is stronger in those towns where initial inequality ($G_{i,2011}$) is high. $\beta$, the marginal effect of town social network fragmentation ($F_i$) on the Gini of the town in 2016 ($G_{i,2016}$), becomes significant around the mean of the Gini in 2011 ($G_{i,2011}$) i.e. at $Z_{G_{i,2011}} = 0$. It increases as $G_{i,2011}$ grows. Points depict estimated marginal effects at the mean and bars represent their 95% confidence intervals ($n = 474$).

town level. The cumulative distribution of income in Ajka (gold) is above the one of Gödöllő (dark blue), indicating lower inequality in the Ajka. We denote the Gini index of town $i$ in year $t$ by $G_{i,t}$, i.e. $G_{\text{Ajka},2011} < G_{\text{Gödöllő},2011}$.

To capture social network structure within towns, we use data retrieved from a Hungarian online social network (OSN) iWiW, a once popular social media platform used by nearly 40% of the country's population. Similar OSN data retrieved from other platforms (e.g. the Dutch OSN Hyves and Facebook) have been used to model income in geographical areas in the Netherlands and in the US[26,27]. In iWiW, we have access to the location of users at the town level and can analyze more than 300 million friendship ties the users have established by the end of 2011.

Previous research demonstrated that geographical factors explain registration rates on the website[48], that administrative and geographical boundaries constitute spatial borders on the iWiW network[49,50] and found relation between network structure indicators and social outcomes such as the prevalence of corruption in towns[51]. A more detailed description of iWiW presented in Supplementary Notes 2 and 3, including its approximation for social and economic representativity and potential biases.

When studying social network fragmentation within a town, we consider only those links between iWiW users that are both from that town. We apply the community detection method known as the Louvain algorithm[52]. This method partitions the

individuals of the network in town $i$ into groups by optimizing a measure called modularity $Q_i$ that compares the density of edges within groups to the density across groups[53]. Mathematically, $Q_i = \sum_{k=1}^{K_i} [\frac{L_k^w}{L_i} - (\frac{L_k}{L_i})^2]$, where $K_i$ is the number of communities for the partition and $L_i$ is the total number of edges in town $i$, $L_k$ is the number of edges adjacent to members of community $k$, and $L_k^w$ is the number of edges within community $k$.

Because $Q_i$ is highly dependent on the size and density of the network, following[54], we scale it by the theoretical $Q_i^{max}$ that would be achieved if all edges were within the communities. The ratio

$$F_i = Q_i / Q_i^{max} \qquad (1)$$

for the town networks provides a good quantitative characterization of their fragmentation[51]. Here we use the values of fragmentation $F_i$ observed for ties created by the end of 2011.

The structure of the social networks in the sample towns Ajka and Gödöllő is illustrated by a filtered, random sample of well-connected nodes from their social networks in Fig. 1. The precise filtering is described in Supplementary Note 2. The network in the relatively unequal town Gödöllő in Fig. 1b is rather fragmented and falls into loosely connected subnetworks compared to the network of Ajka, a town with low inequality, visualized in Fig. 1c. Figure 1d illustrates the positive correlation (Pearson's $\rho = 0.29$ for all towns) between $G_{i,2011}$ and $F_i$ meaning that income inequalities are generally higher in those towns where the social network is fragmented.

Turning to the dynamics of inequality, Fig. 1e illustrates the strong correlation ($\rho = 0.9$) between inequality in 2011 and 2016. We observe a slight increase in the overall level of inequality in most towns from an average Gini index of 0.474 in 2011 to an average of 0.484 in 2016 (significant according to a Mann–Whitney $U$-test, $p < 0.001$). There are examples of towns with both growing and falling inequality.

To analyze how network fragmentation is related to these dynamics, we apply the following ordinary least-squares (OLS) regression:

$$G_{i,2016} = \alpha \times G_{i,2011} + \beta \times F_i + \gamma \times (G_{i,2011} \times F_i) + Z_{i,2011} + \epsilon \qquad (2)$$

where the coefficient $\gamma$ of the interaction term measures the interaction effect of inequality and social network fragmentation. $Z_{i,2011}$ refers to a matrix of control variables (population density and fraction of iWiW users in total population). Here $\beta$ is the regression coefficient for $F_i$ and the total contribution of fragmentation to income inequalities can be estimated from the partial derivative of $G_{i,2016}$ with respect to $F_i$ using the formula

$$\frac{\partial G_{i,2016}}{\partial F_i} = \beta + \gamma \times G_{i,2011}. \qquad (3)$$

Figure 1f presents the relationship between social network fragmentation and the change of town income inequality between 2011 and 2016 and plots $\beta$ by levels of $G_{i,2011}$. We find that the interaction between inequality in 2011 and fragmentation has a positive and statistically significant relationship with inequality in 2016. However, the marginal effect of $F_i$ informs us that social network fragmentation is positively related to future levels of income inequality only if the initial levels of inequalities are high. $F_i$ has no significant relation with $G_{i,2016}$ at the lowest levels of $G_{i,2011}$. One standard deviation change in $F_i$ is associated with 0.1 and 0.4 standard deviation change in $G_{i,2016}$ at the mean and maximum values of $G_{i,2011}$. This result provides empirical support to the theory that social networks can increase inequalities when individuals sort based on their initial endowments[11]. The estimates of Eq. (2) can be found in Supplementary Note 4.

Having established a relationship between social network fragmentation and income inequality, we now turn our attention to potential geographic drivers of such fragmentation, namely the topology of urban space. Divisions or inequalities in geography have long been considered fundamentally related to economic inequality[46,55]. Our goal is to better understand this relationship by observing how geographical factors relate to inequality through their relationship to social networks.

**The role of urban topology.** To test the hypothesis that urban topology is related to income inequality via its relationship to social network fragmentation, we apply a two-stage least square (2SLS) regression model on income inequality. Though we cannot claim that the estimates we derive represent causal effects, our approach minimizes the risk of omitted variable bias[56]. We also exclude a variety of alternative explanations through a series of falsification and robustness tests[46].

In the first stage of the 2SLS model, we estimate social network fragmentation using the formula:

$$F_i = \delta + \gamma IV_i + \delta N_i + e_i \qquad (4)$$

where $IV_i$, short for instrumental variable, denotes our urban topology indicators to be introduced below, $N_i$ is the fraction of the population of a town $i$ on iWiW, and $e_i$ is an error term, assumed to be normally distributed.

The urban structure indicators are created using data from open-source geographic databases. This allows the replication of our measurements in other countries. The geographic features were observed in 2017, lagging our estimates of income inequality and fragmentation. As the geographic features change slowly over the course of many years, the risks of reverse causality are limited. Indeed, many manifestations of segregation and sorting by class were documented in Hungarian towns in the 1970s and 80s[57]. The following indicators are proposed to quantify three dimensions of spatial segregation, the details of which are described in the section on methods.

*Average distance from the center (ADC).* Co-location is important for social tie creation and the probability of links decreases as distance grows[49]. Therefore, large distances between neighborhoods of towns can lead to fragmented social networks, because distant individuals are less likely to meet[26]. However, certain locations in towns facilitate the integration of distant individuals. For example, downtown is assumed to be and indeed functions as the major hub for social interaction in most towns and cities[40]. Therefore, to quantify the role of distance, we measure the average distance of randomly sampled neighborhoods from the center of gravity in towns. Because the form of urban polygons are rarely circular, and may even contain disjoint fragments of neighborhoods that are far from each other (e.g. Esztergom in Fig. 2a) the center of gravity might not coincide with downtown. In these cases the *ADC* index rather captures the large distances across neighborhoods than actual distance from the center. Nevertheless, we expect that social network fragmentation is higher in towns where *ADC* is large and lower where the index value is low.

Although we cannot test a causal effect of *ADC* on social network fragmentation, we do argue that reverse causality is not likely. City growth is a complex phenomenon depending on land use, regulations, economic attractiveness, and transport[58]. Hence, it is not likely that the presence of segregated social groups drives town growth and hence increases distances, especially not in the short or medium term.

*Spatial concentration of amenities (SCA).* Individuals go out and interact in places that are not necessarily located downtown[59].

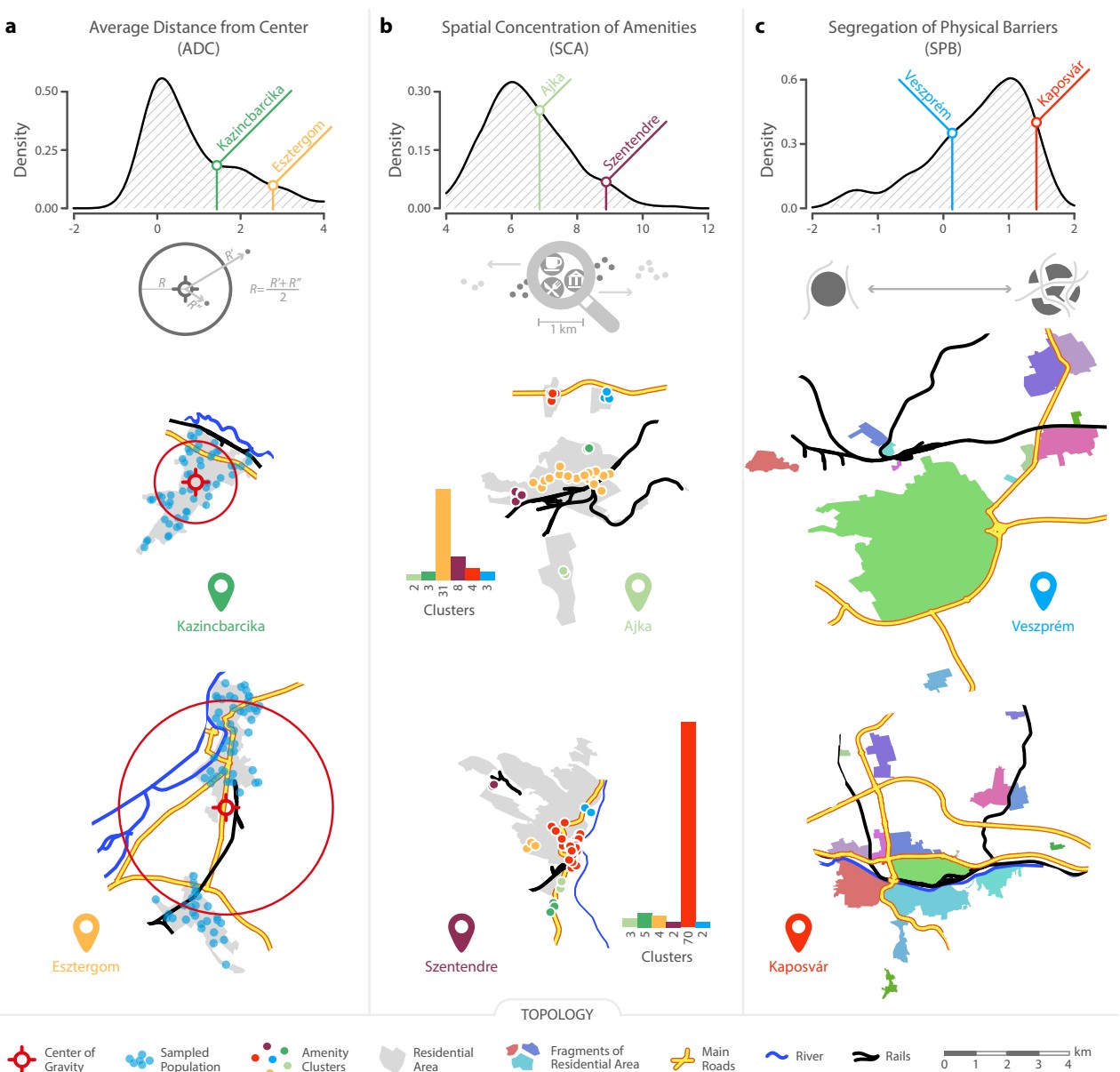

**Fig. 2 Urban topology indicators. a** ADC captures the average distance of neighborhoods from the center of gravity in the town measured in km and scaled by the size of town area. The two example towns both have around 30,000 inhabitants but Kazincbarcika (low ADC) is spatially concentrated while Esztergom (high ADC) has distant fragments of residential zones. **b** SCA measures the concentration of public and private services and amenities in spatial clusters, scaled by the size of town area. The two example towns have similar populations of around 30,000 inhabitants and both of six clusters of amenities. The difference between Ajka (low SCA) and Szentendre (high SCA) is that amenities are strongly concentrated in the largest cluster of Szentendre. **c** SPB quantifies the extent to which residential areas are divided by railways, roads, and rivers into disconnected components, indicated by colors. The two example towns both have around 60,000 inhabitants. Veszprém (low SPB) has a relatively compact structure with few distant residential zones and low degree of disconnection by physical barriers. On the contrary, neighborhoods of Kaposvár (high SPB) are cut into smaller areas by physical barriers.

The spatial concentration of such amenities is related to the location of rich and poor in cities. Unlike in the US, amenities concentrated in the center of European urban areas have been found to attract the rich to and push the poor from central locations[60]. Consequently, when amenities are spatially concentrated, residents in peripheral areas of the town might be excluded from majority of social interaction[61]. However, the question how this concentration is related to social network fragmentation is still open, since amenities distributed evenly across neighborhoods can also facilitate interaction among peripheral neighbors and increase local cohesion to the detriment of overall connectedness in the town[62].

To understand how the *SCA* is related to network fragmentation, we apply point of interest data (POI) that covers restaurants, bars, pharmacies, cinemas etc. To do justice between the two alternative expectations, *SCA* quantifies the concentration of amenities across its spatial groups defined by a density-based clustering algorithm. This measure takes high value if amenities are concentrated in few spatial clusters and low value if they are evenly scattered across spatial clusters (see description in the Methods section).

We find a significant positive correlation with *SCA* and network fragmentation ($\rho = 0.253$), which suggests that network segregation is higher in towns where amenities are spatially concentrated. Therefore, we falsify the alternative hypothesis and

expect a relation between *SCA* and inequalities through network fragmentation. However, we cannot rule out reversed causality, since the location of amenities are based on demand that depend directly on local purchasing power.

*Segregation by physical barriers* (SPB). Both built structures such as major roadways and railroad tracks and natural barriers like rivers are known to facilitate segregation in cities[55]. The effect of such barriers is thought to be an exogenous factor facilitating segregation. Because they can be considered exogenous, they have been used as instrumental variables to measure the effect of racial segregation on disparities in income[46]. This measure encodes the colloquial phenomenon that some neighborhoods are on the "wrong side of the tracks". The effects of physical segregation on socio-economic outcomes are remarkably persistent. For example, neighborhoods in the eastern parts of postindustrial British cities have lower incomes today because they were less desirable places to live in the 19th century when the wind (blowing west to east) concentrated pollution there[63]. Negative externalities such as air pollution are known to cluster in poorer neighborhoods[64]. In our specific context, we expect that social networks are more fragmented in towns that are spatially segregated both because physical barriers decrease the probability of face-to-face interaction[49] and because they facilitate sorting of new arrivals by providing clear boundaries to neighborhoods[65].

Though we cannot demonstrate a causal effect of *SPB* on network fragmentation, reverse causality is unlikely because the barriers are either natural and unchanging or were planned and constructed many years ago. For example, the backbone of Hungarian transportation infrastructure was designed and built in the 19th and early 20th centuries[66], after which very few new railroad tracks have been built. Major new road construction within settlements is rare, and rivers have not been redirected in recent decades.

For the purpose of comparing towns of various sizes, we scale the *ADC*, *SCA*, and *SPB* measures by the total residential area of towns[46]. Figure 2 presents these three measures using three pairs of towns. To illustrate the *ADC* measure we contrast Kazincbarcika, an industrial town in a valley with a concentrated residential area, with Esztergom, a town with two distinct populated areas. Kazincbarcika has a low *ADC* score compared to Esztergom. The *SCA* measure is lower in Ajka, an industrial town that sprawls along a major road, than in Szentendre, a historical town in which amenities (cafes, restaurants, museums, etc.) concentrate in the downtown area. Finally, the *SPB* measure is low in Veszprém: the nearby major roadway wraps around the town and its train station is positioned on the outskirts. In contrast, Kaposvár has a high *SPB* score. The river Kapos and the town's rail link cut the settlement from east to west, while two major roadways intersect near the center of the town.

Results of the estimation specified in Eq. (4) are presented in Table 1 and confirm a significant positive correlation between social network fragmentation $F_i$ and all *ADC*, *SCA*, and *SPB* dimensions of urban segregation. All three indicators capture different facets of potential geographic sources of social segregation. In Supplementary Note 5, we use principal component analysis to construct a composite indicator of urban topology. This composite indicator captures a multidimensional notion of geographic segregation (it is high when all three elementary indicators are high). Description, distribution, and correlation of control variables are described in Supplementary Note 6. Supplementary Note 7 illustrates that all three urban structure indicators and the composite index outperform alternative segregation proxies (measuring ethnic, religious, educational, political heterogeneity) in predicting fragmentation by applying a machine learning approach.

The second stage of the 2SLS estimation follows the formula

$$G_{i,2016} = \alpha + \beta_1 \hat{F}_i + \beta_2 X_i + \varphi_k + e_i \qquad (5)$$

where $\hat{F}_i$ is the predicted value of fragmentation estimated from Equation 4, $\varphi_k$ refers to county-level fixed effects and $e_i$ is the error term. $X_i$ is a matrix of control variables including the level and change of foreign-direct investment, unemployment rate, population density, and distance to the closest border.

Results presented in Table 2 confirm that social network fragmentation, instrumented by urban structure indicators, is positively related with income inequality. This result represents strong evidence of our proposed relationship between social network fragmentation and inequality. It also suggests that urban structure is an important indicator of social network outcomes that coincide with inequality. The control variables suggest that densely populated towns have lower levels of inequality than sparsely populated towns, which is in line with previous findings[67], and inequality in towns close to the border (which tend to be peripheral towns in the case of Hungary) is above average. Supplementary Note 8 reports complete regression tables of the second stages of the 2SLS models.

Model statistics in Table 1 suggest that urban topology indicators are strong instruments of social network fragmentation, confirmed by an *F*-test of the first stage regression. A Wu–Hausman test confirms that they are not significantly correlated with the second stage dependent variable: income inequality. With the exception of the *SCA* regression, the instrumental variable models provide better fit than OLS regressions using the original urban structure measures instead of fragmentation as confirmed by a Sargan test. Robustness checks reported in Supplementary Note 9 confirm that results are mostly stable against restricting the observations to larger towns. Supplementary Note 10 contains further robustness tests, to check whether the effects of urban topology on inequality are mediated by the fragmentation variable, we regress fragmentation and inequality on all explanatory variables from Eqs. (4) and (5). In these comprehensive models we find that SPB is the most robust IV and fragmentation remains a statistically significant predictor of inequality.

It should be emphasized that we cannot prove causal relationships with our modeling approach via robustness and falsification tests. However, comparing the robustness of the instrumental variables allows us to exclude a variety of confounding factors and helps us understand the spatial dimension of social network segregation and its role in income inequality. We also carried out a series of regressions to rule out alternative hypotheses following the falsification strategy of Ananat[46]. In these, we test the correlation of urban topology indicators with other factors that predict town segregation or other dimensions of inequality, for example the level of economic efficiency of a town, measured by business tax receipts (reported in Supplementary Note 11). We find that the *SPB* measure does not correlate with any other proxies of inequality we consider, replicating the previous finding of Ananat that division of physical space by railroads is a sound instrumental variable for the analysis of the effects of segregation on inequality outcomes. *ADC* and to a greater extent *SCA* do correlate with some proxies for inequality, but we note these proxies do not effectively substitute for the urban topology indicators in our primary models. We acknowledge that we cannot exclude all alternative paths of cause and effect between social network fragmentation and economic inequality.

**Table 1 We estimate the relationship between urban topology indicators and social network fragmentation using Eq. (4).**

| | Dependent variable: fragmentation ($F_i$) | | |
| --- | --- | --- | --- |
| | (1) | (2) | (3) |
| ADC | 0.091** (0.045) | | |
| SCA | | 0.110** (0.046) | |
| SPB | | | 0.168*** (0.044) |
| User rate | 0.367*** (0.045) | 0.355*** (0.046) | 0.344*** (0.044) |
| Constant | −0.000 (0.042) | −0.000 (0.042) | −0.000 (0.042) |
| Observations | 473 | 473 | 473 |
| $R^2$ | 0.167 | 0.170 | 0.185 |
| Adjusted $R^2$ | 0.163 | 0.166 | 0.181 |
| Residual Std. Error (df = 470) | 0.915 | 0.913 | 0.905 |
| $F$ Statistic (df = 2; 470) | 47.082*** | 48.085*** | 53.259*** |

We report the first stage of the 2SLS regressions. Standard errors in parentheses; all variables have been standardized.
*$p < 0.1$; **$p < 0.05$; ***$p < 0.01$.

**Table 2 We estimate the relationship between social network fragmentation and income inequality using the urban topology indices as instruments for fragmentation using Eq. (5).**

| | Dependent variable: Gini coefficient ($G_{i,2016}$) | | |
| --- | --- | --- | --- |
| | Instrumental variable | | |
| | ADC | SCA | SPB |
| Estimated fragmentation ($\hat{F}_i$) | 0.408*** (0.153) | 0.533*** (0.146) | 0.288** (0.138) |
| Population density | −0.092* (0.055) | −0.118** (0.052) | −0.067 (0.053) |
| Distance to border | −0.243*** (0.059) | −0.231*** (0.062) | −0.254*** (0.058) |
| Constant | −0.386 (0.369) | −0.392 (0.330) | −0.380 (0.372) |
| County FE | Yes | Yes | Yes |
| Controls | Yes | Yes | Yes |
| First Stage $F$-test | 22.290*** | 24.009*** | 26.754*** |
| Wu–Hausman test | 1.107 | 3.729 | 0.011 |
| Sargan test | 0.051 | 5.349* | 1.400 |
| Observations | 473 | 473 | 473 |
| $R^2$ | 0.231 | 0.192 | 0.245 |
| Adjusted $R^2$ | 0.186 | 0.145 | 0.200 |
| Res.St.Err. (df = 446) | 0.902 | 0.924 | 0.894 |

We report the second stage of the 2SLS regressions. Standard errors in parentheses; all variables have been standardized.
*$p < 0.1$; **$p < 0.05$; ***$p < 0.01$.

## Discussion

In this paper, we demonstrated that the fragmentation of social network structure is positively associated with income inequality in cities and towns. Moreover, we have found that the relationship is dynamic—the interaction of fragmentation and existing inequalities predicts a significant growth in inequality in the future. The physical arrangement of a city's residential areas, the loci of its social interactions, is also connected to social network fragmentation. We observe a tendency: if the urban fabric contains significant distances, physical barriers, or spatially concentrated amenities, social networks tend to be more fragmented. The relationship between geographic division and inequality manifests in this fragmentation.

Our analysis suggests how and why urban planning can be an effective tool to moderate inequalities in the long-run. While it has long been known that segregation is often an implicit goal of urban planning, for example, Detroit's Eight Mile Wall, the barrier that long separated suburban New Haven from Hamden in Connecticut[68], or the widespread phenomenon of gated communities[69], our work suggests that even innocent design choices can lead to bad outcomes. Conversely, certain policies may facilitate mixing and block the compounding of inequality by fragmentation.

For instance, insuring access to places of interaction is known to improve emotional well-being in neighborhoods[70]. Yet evidence for the effectiveness of mixed-income public housing projects in reverting segregation is more limited[71]. Smart policy is especially important when cities are growing: urban sprawl in Beijing has generated significant segregation between economic strata and locals and newcomers[72]. Other work suggests that experimentation and innovation in urban planning is needed to foster the accumulation of good forms of social capital[73].

We describe three urban topology indicators that capture different dimensions of social segregation in cities and can be ordered in terms of changeability. Average Distance from the Centre (ADC), Spatial Concentration of Amenities (SCA), and Segregation of Physical Barriers (SPB) each offer their own insight for policy. Physical barriers is the most robust instrument for social network fragmentation; however, this is the least changeable factor. Its policy implications offer guidance for where to lay railroad tracks and primary roads within cities. In contrast, the distribution of amenities, which can change relatively quickly is a less robust instrument for social network fragmentation. Yet it is more relevant to day-to-day urban planning because public planners make frequent decisions about zoning and building permits for amenities. These might influence the spatial dimension of social interaction and consequently the dynamics of inequalities.

We cannot prove the following story of cause and effect: that poorly designed cities fragment the social network and hence amplify economic inequality. There may be confounding variables that explain our results. Indeed, the long-term evolution of neighborhoods is a complex phenomenon including mechanisms and feedback loops that we can not evaluate in this paper. Nevertheless our observations give us the confidence to propose that the rise of inequalities in towns may be fruitfully blunted by wise urban planning. We hypothesize that improving access across neighborhoods, facilitating mixing within them, and supporting a more equal distribution of services can mend broken social networks and improve economic outcomes across the board.

## Methods

**Materials**. Our data access to the iWiW online social network is controlled by a non-disclosure agreement. The data, besides other information, includes self-reported location of 2.8M users and their social connections reported on the OSN website. Access can be requested in email addressed to: lengyel.balazs@krtk.hu.

Town-level aggregate information including income distributions, population distribution according to school, age, ethnic and religious groups, population density, unemployment, distance from border and foreign-direct investment was collected from https://www.teir.hu.

Corine Land Cover (CLC) data of built up residential areas including continuous and discontinuous urban fabric according to CLC 2012 were collected from the https://land.copernicus.eu/pan-european/corine-land-cover website. Geographic data on the location of residential areas, rivers, railroads, and major roads was collected from Open Street Map https://data2.openstreetmap.hu/hatarok/. Data on POI listed as "amenities" was downloaded using the https://download.geofabrik.de/website.

**Methods**. To calculate *ADC*, we randomly located points on the polygons of residential zones in the CLC database. The number of points in each town refers to its total population and, because we aim to create the measure reflecting on urban topology, the number of points in a polygon is a function of the polygon's area. Based on this randomized spatial distribution, we estimated the center of gravity for the town and calculated the average distance of points from it, following:

$$ADC_i = \frac{\sum_p^P D_{p,c}}{P} / S_i \qquad (6)$$

where $D_{p,c}$ denotes the distance of points $p$ to the estimated center of gravity $c$ out of $P$ points and $S_i$ refers to the size of the town's area. The value of *ADC* is small for compact settlements and is large in towns with remote population fragments.

To calculate *SPB*, we adapt the measure of the physical division of the residential areas of cities known as the Railroad Division Index[46]. We source data on residential-zoned areas in each settlement in the OSM dataset and cut the polygons by the rivers, major roads, and railroads in the settlement. Then, we calculate the resulting dispersion of its residential area across disconnected components:

$$SPB_i = 1 - \sum_a (S_a / S_i)^2 \qquad (7)$$

where $S_i$ refers to the size of the town's area and $S_a$ denotes size of area $a$ after barrier dispersion. The value of *SPB* is small for settlements that are not divided by barriers and large for those where barriers disconnect large fractions of residential areas.

To measure how much spatially concentrated the amenities are in the town, we identify spatial clusters of POI by applying a DBSCAN algorithm with 500 m radius. This algorithm groups those amenities together that are in the close neighborhood of each other. The spatial concentration of the groups is then quantified with the probabilistic entropy of the size distribution of spatial clusters multiplied by minus 1:

$$SCA_i = \frac{\sum_c (p_c \times \log p_c)}{n(c)} / S_i \qquad (8)$$

where $c$ refers to spatial clusters and $p_c$ is the number of POIs in $c$ over the total number of clusters in the town $n(c)$. The value of *SCA* is high for settlements where amenities are concentrated in few spatial clusters and small for those where amenities are evenly.

These calculations have been made in Python (version number 3.7.2).

**Reporting summary**. Further information on research design is available in the Nature Research Reporting Summary linked to this article.

## Data availability

Data tenure was controlled by a non-disclosure agreement between the owner of iWiW data and the research group. Raw data are not publicly available due to privacy considerations, but are available to researchers who meet the criteria for access to confidential data, sign a confidentiality agreement and agree to work under supervision at the Centre for Economic- and Regional Studies. Data access can be requested by email to the corresponding author: lengyel.balazs@krtk.hu. The table that contains town-level variables, can be accessed at https://zenodo.org/record/4448183#.YAXjOOhKg2w. Town-level aggregate socio-economic information can be accessed at https://www.teir.hu. The Corine Land Cover (CLC) data is available at https://land.copernicus.eu/pan-european/corine-land-cover website. The Open Street Map data is accessible at https://data2.openstreetmap.hu/hatarok/and the POI can be downloaded from https://download.geofabrik.de/.

## Code availability

Codes to produce the urban topology indicators, the figures and regression tables can be accessed at https://zenodo.org/record/4448183#.YAXjOOhKg2w.

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

## Acknowledgements

The authors wish to thank Zoltán Elekes, Sándor Juhász, Eszter Bokányi, and László Czaller for their comments and suggestions. We acknowledge the significant assistance of our illustrator Szabolcs Tóth-Zs. in finalizing our primary figures. B.S., J.K., Á.J., and B.L. acknowledge financial support received from National Office for Researcher and Innovation (OTKA K129124). R.D.C. as Newton International Fellow of the Royal Society acknowledges support from the Royal Society, the British Academy, and the Academy of Medical Sciences (Newton International Fellowship, NF170505). G.T. acknowledges the support from the Science Foundation Ireland, SFI Science Policy Research Programme (No. 17/SPR/5324).

## Author contributions

J.W., B.S., J.K. and B.L. designed the research, G.T., J.W., Á.J. and R.D.C. conceived the experiments, G.T., J.W. and B.L. analyzed the results. All authors wrote and reviewed the manuscript.

## Competing interests

The authors declare no competing interests.
