## [Peer Review File · Nature Communications]

Reviewers' Comments:

Reviewer #1:

Remarks to the Author:

- 1) What is the fundamental causal relationship between social network segregation and urban topology? Urban topology cannot be blamed for causing inequality.
- 2) The authors argue "Geography is both an important source and marker of economic inequalities. A stylized fact suggest that home location describes much of individuals' economic potential and access to opportunities through education". Please explain what does geography mean in this paper? Does it only mean location? Does it also consider context, scale, and relative distance? If so, please explain.
- 3) "Yet to our knowledge, big data on social networks have not tested the relationship between social segregation and economic inequality" is a very strong statement. Please explain why. Is that because big data researcher does not know the literature of economic inequality. Or, is there any theoretical gap? Because it is considered by the authors as the major contribution. I hope the authors can clearly explain why other scholars have not explored this topic.
- 4) "In this paper, we analyze a large scale online social network of ca. 2 Million individuals locate in ca." are confusing. The authors need to check the abbreviation.
- 5) "to reduce extreme inequalities by improving the access between neighborhoods" has not been mentioned by [24]. This statement is also wrong. Does it mean the access improvement between rich and poor communities can reduce extreme inequalities? Again, this manuscript has many strong statements declaring that physical configuration can change the social inequality, which is very wrong.
- 6) "What has not yet been quantified is the extent to which urban topology relates to actually observed social segregation" is a very old and outdated idea in urban planning. If it works, we only need architects to build the cities and solve urban problems. The authors might want to argue that spatial network and social network are related. However, it is very challenging to argue that inequality can be driven by spatial configuration.
- 7) The authors argue "geographic constraints are related to economic inequalities". Please sue literature to support the argument. Please note there are many papers about physical access and socioeconomic access.
- 8) "Moreover, unlike social networks, which are difficult to change directly via public policy interventions, cities are significantly shaped by urban planning and policy choices made by governments" is questionable. The authors need to give the evidence and literature support. Most of cities are shaped by history instead of one round of urban planning. In addition, cities are formed by individuals and social networks. Social network cannot change, but cities can change. What does it mean?
- 9) It is ambitious to explore "how the structure of built environment relates to income inequalities through social relations". The authors need to give more in-depth literature review from the perspectives of urban geography, spatial social network, and urban economics. However, the manuscript mainly views cities as the physical system.
- 10) "We find empirical evidence that income inequalities rise more in towns where social networks are fragmented and initial income inequalities are also high" is obvious. Because if a town or community is homogenous, the income inequality is low by definition. A homogenous community tends not to have fragmented social network. In other words, this discovery is actually to prove an existing definition.
- 11) "All 474 Hungarian towns with at least 2500 inhabitants" is questionable. A small town tends to have a lower income inequality due to the homogeneity. Is there any trade-off when putting towns from small size to lager size together?

Reviewer #2:

Remarks to the Author:

This paper, "Inequality is rising where social network segregation interacts with urban topology"

studies the effect of urban topology on town-level income inequality through social network fragmentation, using the large-scale data on tax filings, online social network "friends," and physical urban space in Hungary. The authors found that the modularity of online social networks was correlated with residents' Gini coefficients and also several physical topology indicators of town (such as average distance from center) have a significant impact on the modularity of online social networks in town.

I concur with the authors in the importance of this research subject. The interaction between physical and socioeconomics spaces is an interesting and important subject for study, especially regarding inequality. I am also impressed that the authors collected and combined the various types of data on a nationwide scale. I think, however, that the authors could show results in a more effective way. I am not sure if I correctly understood the figures and table in the paper because of the lack of clarification. Thus, I regret not being able to see whether the analysis properly supports the paper's conclusion with the figures and table.

For example, I was not able to find the explanation about the node color of Fig. 1BC, the actual values of Gini and normalized modularity F for each sample town in Fig. 1BC, the dot color and the line of Fig. 1D, the X and Y axis labels' meaning related to the statistical model (2) of Fig. 1E, and why the kernel density is used instead of dots (like Fig. 1D) of Fig. 1E subgraph. In particular, I was not able to understand what Fig. 1E shows in the statistical model of Eq. (2) and why "This results provide empirical support to the theory that social networks can increase inequalities when individuals sort based on their initial endowments." (p. 4). The table that simply shows the estimated coefficients and their significance test results of Eq. (2) could be helpful to understand.

I faced the same difficulties in, Fig. 2, Fig. 3, and Table 1. It is hard to digest Fig. 2 because there is only partial explanation of the colors and lines in the maps. Also, if the authors showed the actual values of ADC, SPB, and SCA for each town, it would be helpful to see whether the quantities properly represent the physical constraints. I would suggest that the authors keep consistency in wording over the manuscript including figures and table. I could have understood the materials better if they had shown the same variable indicators with the statistical models in the figures and table.

In addition to the materials' clarity, I have concerns about the sampling, the analysis, and the abstract. As for the social network part, the authors used the data of a specific online platform (iWiW) that nearly 30% of people used in Hungary. That is, the social-network data is the sample of each town residents (probably about 30% on average). Because the social media users can be biased regarding income and residential area, it is questionable to apply the characteristics of the biased social network data to the town-wide parameters such as income inequality and physical topology indicators. For example, if the most social media users were categorized to the low-income category, it would be less meaningful to compare the data with the Gini coefficient of entire residents. I would suggest that the authors show the evidence that the social media users are not significantly biased in the residents of each town. Also, the authors used the social network data until 2011 to evaluate the Gini coefficient in 2016. I think they need more justification for the analysis.

As for the analysis, the authors used principal component analysis (PCA) to make one physical topology indicator (CUTI) from the three separate ones (ADC, SPB, and SCA). But they also used the each separate indicator for the statistical analysis of Eq. (3) (Fig. 3) and that of Eq. (4) (Table 1). I think it violates the assumption of PCA. The indicator combined by PCA is meaningful when the three separate indicators share the same component. First, to clarify this point, the authors should show the PCA result. Second, they should select whether they use the single indicator CUTI or the separate three ones for the main analysis. If they think the separate analyses using the three indicators are meaningful, that means they assume that these separate indicators are independent and have a different impact on social network fragmentation; in that case, the combined indicator CUTI would be meaningless.

Finally, I found a logical jump in abstract. The abstract's 3rd sentence says, "Yet, little is known about what structural factors facilitate fragmentation.", which I think correspond to how ADC, SPB, and SCA affect the normalized modularity of online social networks F in the paper. Then, the 4th sentence says, "there is a significant relationship between social network fragmentation and income inequality in cities and towns." which I think are the normalized modularity F and the Gini of residents' income G . There is a logical jump between the two sentences, and the authors should fill up the gap. Also, the 5th sentence mentions the comparison with "unequal access to education, political segregation, or the presence of ethnic and religious minorities.", but I was not able to find a clear result about this point in the manuscript.

Response to Referees' Comments

First of all, we would like to thank both referees for their critical but useful and constructive remarks. We have addressed all of them, which resulted in a major revision and improved the quality of our paper.

We have rewritten most of the Introduction and Discussion sections including more literature and formulating a better, clearer line of argument. Figure 1 and Figure 2 have been improved and the communication of the two-stage least-squares (2SLS) regression analysis has been made more clear. Additional Supporting Information sections describe potential biases of iWiW representativity, robustness against restricting the sample to larger towns, etc.

In the following, we address comments one by one following each comment.

Thank you once again for your work.

Referee 1:

Thank you for your critical but very constructive remarks. These have helped us to reframe the motivation of the paper.

1) What is the fundamental causal relationship between social network segregation and urban topology? Urban topology cannot be blamed for causing inequality.

Answer: Accepting your critique, we have emphasized statistical significance of the apparent relationship between urban topology and inequality. Moreover, by engaging with the urban economics and sociology literature, we paint a more general picture on how physical geography (captured by urban topology indicators), social network fragmentation and economic inequality are jointly related.

Let us describe why we think this relationship is important to quantify. We know from the network science literature that geographical barriers such as distance and physical or administrative boundaries fragment country-scale social networks into spatial communities (Expert et al. 2011, PNAS; Lambiotte et al. 2008, Physica A; Lengyel et al. 2015, PLoS ONE). The spatial fragmentation of social networks has been documented within urban areas as well (Bailey et al. 2018, Journal of Economic Perspectives). We also know from the sociology literature that fragmented social networks amplify inequalities in case individual wealth and economic potentials play some role in how social relations are established (DiMaggio and Garip 2012, Annual Review of Sociology). It is not clear however, whether the structure of geographically embedded social networks function as a link between physical geography and the unequal distribution of wealth.

Our paper addresses this issue. We use the town-level variation of urban topology, social network fragmentation and income distribution to address the above research gap. The introduction has been rewritten accordingly.

2) The authors argue “Geography is both an important source and marker of economic inequalities. A stylized fact suggest that home location describes much of individuals' economic potential and access to opportunities through education”. Please explain what does geography mean in this paper? Does it only mean location? Does it also consider context, scale, and relative distance? If so, please explain.

Answer: Thank you for this comment. In this paper, we take an approach different from most quantitative work on spatial economic output. In our view, geography not only refers to the location of individuals but also plays a fundamental role in enabling or hindering social relationships.

A significant body of literature suggests that physical and administrative barriers on country scale (such as rivers, mountains or borders) and physical barriers on towns scale (such as major roads, tracks, and rivers) are obstacles to social interaction beyond what might be caused by increases in relative distances. The history of cities contains many examples of the use of specific barriers as landmarks delimiting neighborhoods, for a review see Schindler (2014). These landmarks facilitate discrimination (for example the practice of redlining) and the unequal provision of public goods, see Noonan (2005). Over time these differences compound because clearly defined neighborhoods facilitate sorting by newcomers.

3) “Yet to our knowledge, big data on social networks have not tested the relationship between social segregation and economic inequality” is a very strong statement. Please explain why. Is that because big data researcher does not know the literature of economic inequality. Or, is there any theoretical gap? Because it is considered by the authors as the major contribution. I hope the authors can clearly explain why other scholars have not explored this topic.

Answer: We think that the lack of research in this domain is due to the difficulty of finding data that captures both the universe of social interactions through large-scale social networks and also income distributions, rather than any lack of theoretical interest. Papers that have access to both have so far primarily focused on economic growth of cities (Norbutas and Corten 2018, Social Networks), on individual wealth (Eagle et al. 2010, Science) or average income in regions (Bailey et al. 2018, Journal of Economic Perspectives).

Your comment informed us that we have to be more specific in claiming our research’s focus. We have done this, as described in our response to Point 1.

4) “In this paper, we analyze a large scale online social network of ca. 2 Million individuals locate in ca.” are confusing. The authors need to check the abbreviation.

Answer: Thank you for pointing this out. We have edited the relevant paragraph.

5) “to reduce extreme inequalities by improving the access between neighborhoods” has not been mentioned by [24]. This statement is also wrong. Does it mean the access improvement between rich and poor communities can reduce extreme inequalities? Again, this manuscript has many strong statements declaring that physical configuration can change the social inequality, which is very wrong.

Answer: We acknowledge and accept this critique, and so we have rewritten this part of the Introduction by focusing on what the urban economics and urban sociology literatures tell us about the relation between urban form, social networks and segregation. We provide more details in Point 9, which is closely related.

6) “What has not yet been quantified is the extent to which urban topology relates to actually observed social segregation” is a very old and outdated idea in urban planning. If it works, we only need architects to build the cities and solve urban problems. The authors might want to argue that spatial network and social network are related. However, it is very challenging to argue that inequality can be driven by spatial configuration.

Answer: Structural effects are neither deterministic nor irrelevant. Certainly, inequality is not just a result of planning, but neither are the two independent. We agree that physical geography itself cannot be blamed for causing inequality. Architects and planners are not able to solve urban problems alone and we acknowledge this in a more fundamental way in the revised manuscript. However, we do stand by our claim that the structure of cities can foster social mechanisms that lead to unequal development, and that studying the relationships between structure and outcome can provide actionable recommendations to planners.

7) The authors argue “geographic constraints are related to economic inequalities”. Please sue literature to support the argument. Please note there are many papers about physical access and socioeconomic access.

Answer: This point of the argument has been put more to the front, where we argue that geographically constrained social networks are limiting economic potentials. Several convincing papers look at geographic neighborhoods and inequalities in digital services like uber/ubereats: <https://dl.acm.org/doi/abs/10.1145/2675133.2675278>, and at the practice of redlining: <https://journals.sagepub.com/doi/pdf/10.1080/00420980500231720>.

8) “Moreover, unlike social networks, which are difficult to change directly via public policy interventions, cities are significantly shaped by urban planning and policy choices made by governments” is questionable. The authors need to give the evidence and literature support. Most of cities are shaped by history instead of one round of urban planning. In addition, cities are formed by individuals and social networks. Social network cannot change, but cities can change. What does it mean?

Answer: Accepting this critique, we have deleted this sentence from the introduction and revised some points that are related to urban planning in the Discussion of the paper.

9) It is ambitious to explore “how the structure of built environment relates to income inequalities through social relations”. The authors need to give more in-depth literature review from the perspectives of urban geography, spatial social network, and urban economics. However, the manuscript mainly views cities as the physical system.

Answer: We agree that this is a fascinating and wide-ranging question that we are only taking first steps to addressing. Accepting your remark, we have included a more in-depth literature review on urban economics and urban sociology literatures into the Introduction. We have found that previous research in these domains had dealt with the complex relation between inequality, social networks and urban geography.

In urban economics, theoretical discussion revolves around how social relations influence the location decision of agents and how these are related to segregation.

Urban sociology stresses that spatial separation by administrative or physical barriers force social segregation. It is also claimed that the emergent islands of segregation are difficult to bridge by social ties because poor and rich neighborhoods are typically located far from each other.

Several of these papers theorize that urban planning has a mediating role in the relationship between social networks and inequalities, referring to concepts including access to services and physical barriers within the environment. We engage with this literature in our discussion.

10) “We find empirical evidence that income inequalities rise more in towns where social networks are fragmented and initial income inequalities are also high” is obvious. Because if a town or community is homogenous, the income inequality is low by definition. A homogenous community tends not to have fragmented social network. In other words, this discovery is actually to prove an existing definition.

Answer: It is not clear why Referee 1 thinks that homogeneity of the income distribution has a deterministic relation with social network structure. There was no reference included in the comment, which makes the argument of Referee 1 difficult to follow. Social networks in towns might be indeed influenced by income distribution such that the rich are not connected to the poor. However, one can envisage several other situations in which income distribution is homogenous (eg. everyone is rich or everyone is poor) but the social network is fragmented by a variety of other dimensions (eg. the group who attend basketball matches is not connected to those who attend handball matches).

In their review on inequalities and social networks, Dimaggio and Garip (2012, Annual Review of Sociology) stress that groups in social networks diverge in terms of economic status only if the base of social tie formation is economic status.

In Figure 1, we document this relation for large communities such as towns for the first time. First, we show that income inequality is indeed correlated with network fragmentation. Then, by plotting

the marginal effect of network segregation on future levels of income inequality, we illustrate that network fragmentation is positively correlated to income inequalities only in those towns where initial inequality was high as well. This is in line with Dimaggio and Garip (2012) and with the mainstream view on the role of social networks in producing inequalities. However, documenting this finding is still an important contribution. It is also important to demonstrate this relationship to lend greater credibility to our subsequent analyses.

11) “All 474 Hungarian towns with at least 2500 inhabitants” is questionable. A small town tends to have a lower income inequality due to the homogeneity. Is there any trade-off when putting towns from small size to larger size together?

Answer: Thank you for this question. The reason to include small towns is to increase the number of observations in the statistical analysis.

The threat of putting together small towns with large ones hence comparing diverse city systems with small village communities has been handled with a set of techniques in the paper. First, we use the normalized version of the Gini coefficient, thus our measure of income inequality is not sensitive to population size but captures how incomes vary relative to other citizens in the town. Indeed, the Gini coefficient only marginally correlates with town population ($\rho=0.13$). Second, we apply a measure of social network segregation that has been designed to be compared across populations of different sizes, see the discussion in the Methods section. This measure is only slightly correlated with population ($\rho=0.23$) compared to the strong correlation between the raw value of network modularity and population (see inserted figure below). However, to be on the safe side, we control for the share of users among the total population in the town when regressing fragmentation on urban topology indicators and population density in the second stage. Third, urban topology indicators have been scaled with area size of towns.

Driven by your critique, we have conducted further robustness checks by limiting the sample to towns that have (1) more than 3000 inhabitants and then (2) more than 5000 inhabitants in the 2SLS regression. Results are reported in Supporting Information 7. In general, correlations

between urban topology indicators remain significant and even gain more strength when limiting the sample to larger towns. The only exception is SPB that loses significance if towns size is above 5000 inhabitants.

Referee 2

Thank you for your very helpful comments and suggestions. We have closely followed them to improve the communication of our results. We have cut up your review into what we thought were reasonable breaks.

a. This paper, "Inequality is rising where social network segregation interacts with urban topology" studies the effect of urban topology on town-level income inequality through social network fragmentation, using the large-scale data on tax filings, online social network "friends," and physical urban space in Hungary. The authors found that the modularity of online social networks was correlated with residents' Gini coefficients and also several physical topology indicators of town (such as average distance from center) have a significant impact on the modularity of online social networks in town.

I concur with the authors in the importance of this research subject. The interaction between physical and socioeconomic spaces is an interesting and important subject for study, especially regarding inequality. I am also impressed that the authors collected and combined the various types of data on a nationwide scale. I think, however, that the authors could show results in a more effective way. I am not sure if I correctly understood the figures and table in the paper because of the lack of clarification. Thus, I regret not being able to see whether the analysis properly supports the paper's conclusion with the figures and table.

Answer: Thank you for this nice summary and the important suggestion. We have worked on the figures and tables and have rewritten their captions and descriptions in the text.

b. For example, I was not able to find the explanation about the node color of Fig. 1BC, the actual values of Gini and normalized modularity F for each sample town in Fig. 1BC, the dot color and the line of Fig. 1D, the X and Y axis labels' meaning related to the statistical model (2) of Fig. 1E, and why the kernel density is used instead of dots (like Fig. 1D) of Fig. 1E subgraph. In particular, I was not able to understand what Fig. 1E shows in the statistical model of Eq. (2) and why "This results provide empirical support to the theory that social networks can increase inequalities when individuals sort based on their initial endowments." (p. 4). The table that simply shows the estimated coefficients and their significance test results of Eq. (2) could be helpful to understand.

Answer: We have improved Figure 1 along these guidelines. Colors in the network plots are now explained; Gini and Fragmentation levels are reported, the line is explained. We have made axis labels and statistical models consistent. We suggest that the kernel density plot is more appropriate than a usual scatter plot in this situation because it enables us to see where the observations are concentrated.

Fig 1E has been improved. We have made further efforts to clarify its meaning with better axis labels, more self-explaining visualization and better link to the text. We think it is still better to include this figure in the main text instead of a table because a simple interaction coefficient could not inform us about the differences of F influence given different values of G but captures joint effects at the average of the variables. We put the regression table that you requested into an SI.

- c. I faced the same difficulties in, Fig. 2, Fig. 3, and Table 1. It is hard to digest Fig. 2 because there is only partial explanation of the colors and lines in the maps. Also, if the authors showed the actual values of ADC, SPB, and SCA for each town, it would be helpful to see whether the quantities properly represent the physical constraints. I would suggest that the authors keep consistency in wording over the manuscript including figures and table. I could have understood the materials better if they had shown the same variable indicators with the statistical models in the figures and table.

Answer: In the new version of Figure 2, we include the variable distributions and denote the sample towns there. Wording in variable indicators have been made consistent.

- d. In addition to the materials' clarity, I have concerns about the sampling, the analysis, and the abstract. As for the social network part, the authors used the data of a specific online platform (iWiW) that nearly 30% of people used in Hungary. That is, the social-network data is the sample of each town residents (probably about 30% on average). Because the social media users can be biased regarding income and residential area, it is questionable to apply the characteristics of the biased social network data to the town-wide parameters such as income inequality and physical topology indicators. For example, if the most social media users were categorized to the low-income category, it would be less meaningful to compare the data with the Gini coefficient of entire residents. I would suggest that the authors show the evidence that the social media users are not significantly biased in the residents of each town. Also, the authors used the social network data until 2011 to evaluate the Gini coefficient in 2016. I think they need more justification for the analysis.

Answer: Thank you for raising this point. By definition, iWiW could have been adopted by those who were online and the poorest people were most likely not on iWiW. Previous research found that income per capita in towns indeed correlates positively with iWiW user rate. We report this in a Supporting Information section and argue that the most segregated groups are those, who were not on iWiW. Thus, the bias towards the rich makes our statistical models underestimate the role of social network segregation.

Nevertheless, we control for iWiW use rate in our analyses. We are also confident about the robustness of our results to this bias because we find a significant interaction effect in our model including a lagged term of the dependent variable.

- e. As for the analysis, the authors used principal component analysis (PCA) to make one physical topology indicator (CUTI) from the three separate ones (ADC, SPB, and SCA). But they also used the each separate indicator for the statistical analysis of Eq. (3) (Fig. 3) and that of Eq. (4) (Table 1). I think it violates the assumption of PCA. The indicator

combined by PCA is meaningful when the three separate indicators share the same component. First, to clarify this point, the authors should show the PCA result. Second, they should select whether they use the single indicator CUTI or the separate three ones for the main analysis. If they think the separate analyses using the three indicators are meaningful, that means they assume that these separate indicators are independent and have a different impact on social network fragmentation; in that case, the combined indicator CUTI would be meaningless.

Answer: For the sake of simplicity and exposition we have chosen to remove the composite indicator from the main text.

- f. Finally, I found a logical jump in abstract. The abstract's 3rd sentence says, "Yet, little is known about what structural factors facilitate fragmentation.", which I think correspond to how ADC, SPB, and SCA affect the normalized modularity of online social networks F in the paper. Then, the 4th sentence says, "there is a significant relationship between social network fragmentation and income inequality in cities and towns." which I think are the normalized modularity F and the Gini of residents' income G . There is a logical jump between the two sentences, and the authors should fill up the gap. Also, the 5th sentence mentions the comparison with "unequal access to education, political segregation, or the presence of ethnic and religious minorities.", but I was not able to find a clear result about this point in the manuscript.

Answer: Thank you for this very helpful comment. We have rewritten the abstract to address this. We have introduced our take on the state-of-the-art of related research in the first three sentences in order to better acclimate the reader to our context. This is followed by a more precise research niche description in the fourth sentence. The discussion of our findings are simplified in the remainder of the abstract.

The exercise, in which we compare the role of urban topology in fragmenting networks with other demographic and social factors is included in Supporting Information 5 because this would break the line of argument in the main text. We included this sentence in the abstract to stress the importance of urban topology compared to town demography. Acknowledging that such details are not necessary in the abstract, we have simplified the communication of the findings there and left a bit more detailed notes in the introduction.

Reviewers' Comments:

Reviewer #2:

Remarks to the Author:

I appreciate the authors' revision that addressed many of my previous comments. I am also satisfied with their elaborate description about the details of the OSN data with demographics. With the supplementary information, I was able to understand the possible limitations and bias of the OSN data. However, I am concerned about the conclusion with the analysis (partially because now I can understand the analytical structure better). Also, I still cannot fully understand the figures, especially Figure 2. The authors could do better to improve the clarity of the figures.

The main conclusion of this paper is "urban topology has a significant relationship with income inequality via social network fragmentation." (from the abstract). The authors show with the two-stage OLS that each factor of urban topology (each of ADC, SCA, and SPB) significantly affects the fragmentation metric of online social networks and that the estimated fragmentation metric significantly affects the Gini coefficient as of 2016. First, it is essential to put events in chronological order for this type of causal inference. However, I couldn't find the information about what point in time the network data and urban topology was from. For example, if the urban topology were as of 2016 and the social network data were as of 2013, the first OLS would be less meaningful. The same concern is true with the second OLS. Second, if the authors argue that social network fragmentation plays a key role in the influence from urban topology to income inequality, they should test, with a comprehensive model including all the independent variables of Table 1 and 2, whether the effects of urban topology on Gini coefficient are mediated by the network fragmentation variable. Because there are other possible intervening variables such as economic efficiency, I don't think that the two-stage OLS alone provides sufficient evidence that urban topology influences income inequality THROUGH social network fragmentation.

The authors updated the figures to improve their clarity. I think, however, there is still room for improvement especially in Figure 2. First, I'm not sure if the two towns Kaposvar and Veszprem are the good samples to represent urban topology indicators because their ADC and SCA are similar. It's hard to see what aspect of urban topology ADC and SCA discern with the visual comparison of the two towns. Second, there are still unclear indications in the figure. What do the purple circles and red dots mean in the ADC column? How are the "amenities" represented in the SCA column? What does "Cluster 1, 2, ..." mean in the SCA column? What does the color of area mean in the SPB column? The authors use pink color both for the city center and for the area segregation. It is confusing. Finally, the authors use different diminishing scale for Kaposvar and Veszprem. It is problematic especially because Figure 2 aims to show space information. We cannot compare distance, spatial concentration, and spatial segregation on different scale.

Specific comments:

1. The authors name the two samples "low income inequality" and "high income inequality" in Figure 1. But they are actually Ajka and Godollo. The authors should include the actual town name for the indicators. Otherwise, Figure 1A shows a kind of tautology (e.g., the curves show the high inequality of "high inequality").
2. The x axis of Figure 1A is not linear. It can give readers misimpression. The authors should follow the standard practice that uses the fraction of population for the x axis.
3. In the caption of Figure 1, "Node colors present communities." It is unclear. What do the community mean here? How do the authors cluster those people? Network-based? Or based on the sub living area of each town?
4. In Figure 1E, the authors indicate the number of single pair of CI (0.258 – 0.702). But there are many bars in the graph. What does the CI values represent?
5. The author should indicate G_i , 2016 instead of G_i in Equation 5. Since the authors has two Gini coefficients with different time period, they should clarify which is the dependent variable in Equation 5 (and other equations).

6. In Table 2, "Fragmentation (Fi)" should be "Estimated fragmentation (circumflex Fi)". The difference is critical here.

Rebuttal Letter

We would like to thank the editor and reviewer for giving us another opportunity to revise and improve our manuscript. We have extended our analyses with several additional robustness checks, to address concerns raised by the reviewer. Additionally we have revamped our primary figures and the accompanying exposition to improve clarity. We feel the resulting manuscript improves on the previous draft substantially. We address the reviewer directly, responding to points raised (in **bold**) in turn (*italics*).

Reviewer #2 (Remarks to the Author):

I appreciate the authors' revision that addressed many of my previous comments. I am also satisfied with their elaborate description about the details of the OSN data with demographics. With the supplementary information, I was able to understand the possible limitations and bias of the OSN data. However, I am concerned about the conclusion with the analysis (partially because now I can understand the analytical structure better). Also, I still cannot fully understand the figures, especially Figure 2. The authors could do better to improve the clarity of the figures.

We thank the reviewer for their continued engagement.

The main conclusion of this paper is “urban topology has a significant relationship with income inequality via social network fragmentation.” (from the abstract). The authors show with the two-stage OLS that each factor of urban topology (each of ADC, SCA, and SPB) significantly affects the fragmentation metric of online social networks and that the estimated fragmentation metric significantly affects the Gini coefficient as of 2016. First, it is essential to put events in chronological order for this type of causal inference. However, I couldn't find the information about what point in time the network data and urban topology was from. For example, if the urban topology were as of 2016 and the social network data were as of 2013, the first OLS would be less meaningful. The same concern is true with the second OLS.

We thank the reviewer for raising this important and valid concern. We have updated the manuscript to note that the social network data includes connections between users made by the end of 2011, and that our measures of urban topology come from 2018. Though repeated observations of these key variables at times matching our inequality measures would be ideal, we believe that the snapshots we use are appropriate for the inferences we make.

Though our results represent convincing evidence of a relationship between network fragmentation on inequalities, the reviewer is certainly correct that we cannot prove causality. Therefore, we only discuss correlations between urban topology and social network structure, that is based on the role of distance and spatial barriers that have been frequently reported in previous literature (Liben-Novel et al. PNAS 2005, Expert et al. PNAS 2011).

More concretely, in this comment, the reviewer raises concerns about reverse causality. The information we use for creating the urban topology indices are arguably stable over time. Rails, roads, and rivers change very rarely, and not since 50 years in almost all cases that makes us think that the SPB indicator should be stable. Town size also changes slowly; in our case the

country has had a shrinking population for 30 years. Single amenities can appear or disappear more quickly, though their distribution should not change substantially over 10 years.

To bolster our case that our geographic measures evolve on a slower time scale, we present additional evidence in this reply. Specifically, we compared data on the number of restaurants and bars in each town in our sample from the Hungarian Statistical Office in 2000 and 2010. Among the 341 towns (out of 473 analyzed in the paper) for which data is available in both years, the Pearson correlation in the (log) number of amenities present is .96. This suggests that even our most temporally volatile geographic measure changes quite slowly.

Second, if the authors argue that social network fragmentation plays a key role in the influence from urban topology to income inequality, they should test, with a comprehensive model including all the independent variables of Table 1 and 2, whether the effects of urban topology on Gini coefficient are mediated by the network fragmentation variable. Because there are other possible intervening variables such as economic efficiency, I don't think that the two-stage OLS alone provides sufficient evidence that urban topology influences income inequality THROUGH social network fragmentation.

The reviewer is correct that it is difficult to prove the path of relationships. What we can do is falsify alternative explanations. We do this in two ways: testing a comprehensive model and carrying out a battery of falsification tests, imitating Ananat (AEJ 2011), who uses a similar railroad division index as an instrument to estimate the effects of segregation on racial income inequality.

First, following the suggestion of the reviewer, we have tested the comprehensive model in which we included all variables from Table 1 and Table 2. Specifically, an OLS model predicting town Gini in 2016 with social fragmentation, the three urban geography indicators, and controls shows that fragmentation has the strongest relationship with the dependent variable. This model is reported in Supporting Information 10.

Second, following Ananat we have carried out falsification tests to rule out counterfactual explanations, reported in SI 11. To that end, we run 12 robustness regressions for each of ADC, SCA, and SPB separately. The first four test the relationship of the indicator with alternative measures of social fragmentation (social network fragmentation, townsize, ethnic fractionalization, and share of the population that is Roma). The remaining 8 regressions test the relationship between the geographic indicators and socio-economic correlates of income inequality. Significant relationships would highlight potential alternative paths of cause and effect that our primary specification cannot rule out. All regressions include controls and county fixed effects (imitating our main analysis).

We find that SPB has a significant relationship only with our social network measure of fragmentation (at $p < .05$). It has no significant relationships with proxies for social fragmentation or economic inequality. Encouragingly, our indicator that most closely resembles the measure of Ananat directly replicates her findings.

ADC has a significant relationship with town size ($p < .001$) and the share of high school graduates in a town ($p < .05$). In the other 10 regressions there are no significant relationships.

SCA has a significant relationship with two of the four proxies of social fragmentation and three of the eight proxies for economic inequality. SCA seems to be the least robust indicator according to these tests. We note that these robustness tests are rather strict because we do not make multiple-hypothesis test corrections to the p -values. We also note that the proxies do not substitute for the geographic indicators in our primary specification. Nevertheless our revised text acknowledges these limitations.

The authors updated the figures to improve their clarity. I think, however, there is still room for improvement especially in Figure 2. First, I'm not sure if the two towns Kaposvar and Veszprem are the good samples to represent urban topology indicators because their ADC and SCA are similar. It's hard to see what aspect of urban topology ADC and SCA discern with the visual comparison of the two towns.

Thank you for raising this issue. We have replaced Veszprém and Kaposvár with two pairs of towns with similar size to illustrate high and low values of ADC and SCA indicators.

Second, there are still unclear indications in the figure. What do the purple circles and red dots mean in the ADC column? How are the “amenities” represented in the SCA column? What does “Cluster 1, 2, ...” mean in the SCA column? What does the color of area mean in the SPB column? The authors use pink color both for the city center and for the area segregation. It is confusing. Finally, the authors use different diminishing scale for Kaposvar and Veszprem. It is problematic especially because Figure 2 aims to show space information. We cannot compare distance, spatial concentration, and spatial segregation on different scale.

Thank you for the specific comments regarding Figure 2. We have carefully redrawn both Figure 1 and Figure 2. We believe that the new versions establish clearer messages and include all important information. We included a detailed legend at the bottom of the figure to explain all notations and an iconographic representation of the variables. The scale of the maps has been made uniform. We have also revised the captions.

Specific comments:

1. The authors name the two samples “low income inequality” and “high income inequality” in Figure 1. But they are actually Ajka and Godollo. The authors should include the actual town name for the indicators. Otherwise, Figure 1A shows a kind of tautology (e.g., the curves show the high inequality of “high inequality”).

Thank you for the comment, we have followed your suggestion and have redrawn Figure 1A.

2. The x axis of Figure 1A is not linear. It can give readers misimpression. The authors should follow the standard practice that uses the fraction of population for the x axis.

We agree that linear income categories would be ideal for illustrating and calculating income inequalities in towns. Unfortunately, we have to follow the income categories of the Hungarian Statistical Office that is the only available income data for towns in this decade besides year 2011 that is a census year and we could potentially use individual-level income. Therefore, we have decided to use the income categories that enables us to compare inequalities across years.

The data include total income and total population by income categories. A standard Lorenz curve drawn from this granularity would be misleading since population fractions (eg. deciles) do not follow the categories and therefore income levels assigned to them would be arbitrary. Accepting your critique and appreciating your helpful comment, we still opt for the raw accumulation of income along the categories we have, which might be less elegant, but it is correct and can inform the reader about the data we analyze.

We document the database in the Materials and Methods section and Gini calculation from income categories in Supporting Information 1.

In this revision, we better explain in the main text and also in the caption of Figure 1 A that cumulative income is based on income categories and in the caption of Figure 1D that Gini is calculated from these categories.

3. In the caption of Figure 1, “Node colors present communities.” It is unclear. What do the community mean here? How do the authors cluster those people? Network-based? Or based on the sub living area of each town?

Thank you for the comment. Communities mean network communities revealed by the Louvain method from the social network within the town. We make this clear in the new caption of Figure 1 B and C.

4. In Figure 1E, the authors indicate the number of single pair of CI (0.258 – 0.702). But there are many bars in the graph. What does the CI values represent?

Thank you for the comment. The text has been removed from the figure.

5. The author should indicate G_i , 2016 instead of G_i in Equation 5. Since the authors has two Gini coefficients with different time period, they should clarify which is the dependent variable in Equation 5 (and other equations).

Thank you, we have corrected Equation 5.

**6. In Table 2, “Fragmentation (Fi)” should be “Estimated fragmentation (circumflex Fi)”.
The difference is critical here.**

Thank you for the note. We have corrected Table 2 accordingly.

Reviewers' Comments:

Reviewer #2:

Remarks to the Author:

I appreciate the significant improvement made by the authors on both the materials and the data interpretations. Figures 1 and 2 that I had clarification concerns in the last review now provide clear information on the relationships between wealth inequality and urban topology via social network fragmentation. I think that the current version of this paper will give a great contribution to our understanding of the roles of urban and social network topologies in economic development. I would recommend this paper for publication by Nature Communications possibly with minor changes to resolve the following minor concerns. I would be grateful, if the authors would have a chance to consider whether they could make the paper better.

Specific comments:

1. I appreciate the comprehensive model of Supporting Information 9. As the authors argue, the result clearly shows the mediation by social network fragmentation from the influence path from urban topology to economic inequality. However, I am still not sure why the authors separately examine the effects of urban topology variables, ADC, SCA, and SPB on the network fragmentation variable F in Table 1 (and as a result, the analyses of Table 2). The urban topologies can be correlated with each other, and the one can be a confounding factor in the relationship between the other variables and network fragmentation. When we included all the three variables in the regression model with the dependent variable of F, we could address the concern. Nevertheless, the authors use the separate statistical models in terms of the urban topology variables. I would suggest that the authors elaborate more on this modeling choice.
2. This paper examines the interaction between social and physical spaces at the city level. Thus, it can help readers to understand the term "fragmentation" that the authors make a clearer distinction between them. For example, they put "Fragmentation" in the x-axis of Fig. 1D, but it's not immediately obvious that it means the "(online) social-network" fragmentation and not "residential" fragmentation because of this study's nature. I think that the authors use "fragmentation" only for social space, but it might be good to add some adjective to the term.
3. In the last paragraph of page 2, the authors mention that "The capital Budapest, which is an order of magnitude larger than the second largest city, is excluded from the analysis." I would suggest that the authors would elaborate a little more on the appropriateness of the data exclusion.